# Exome Analysis Reveals Novel Missense and Deletion Variants in the *CC2D2A* Gene as Causative of Joubert Syndrome

**DOI:** 10.3390/genes14040810

**Published:** 2023-03-28

**Authors:** Rute Luísa Cabrita Pinto, Silvia Viaggi, Edoardo Canale, Marina Martinez Popple, Valeria Capra, Giuseppina Conteduca, Barbara Testa, Domenico Coviello, Angela Elvira Covone

**Affiliations:** 1Laboratory of Human Genetics, IRCCS Istituto Giannina Gaslini, 16147 Genoa, Italy; 2Department of Earth, Environmental and Life Sciences, University of Genoa, 16132 Genoa, Italy; 3Infantile Neuropsychiatry Unit, IRCCS Istituto Giannina Gaslini, 16147 Genoa, Italy; 4Medical Genetics Unit, IRCCS Istituto Giannina Gaslini, 16147 Genoa, Italy

**Keywords:** *CC2D2A*, compound heterozygous, deletion, Joubert Syndrome, missense, pediatric

## Abstract

The *CC2D2A* gene is essential for primary cilia formation, and its disruption has been associated with Joubert Syndrome-9 (JBTS9), a ciliopathy with typical neurodevelopmental features. Here, we describe an Italian pediatric patient with typical features of Joubert Syndrome (JBTS): “Molar Tooth Sign”, global developmental delay, nystagmus, mild hypotonia, and oculomotor apraxia. Whole exome sequencing and segregation analysis identified in our infant patient a novel heterozygous germline missense variant c.3626C > T; p.(Pro1209Leu) inherited from the father and a novel 7.16 kb deletion inherited from the mother. To the best of our knowledge, this is the first report showing a novel missense and deletion variant involving exon 30 of the *CC2D2A* gene.

## 1. Introduction

The rare complex neurological condition known as Joubert Syndrome (JBTS) is caused by pathogenic variants in genes codifying primary cilium proteins, which categorize JBTS in the large group of ciliopathies [1]. Epidemiological data [2] suggests JBTS prevalence to be between 1/80,000 and 1/100,000, and even higher in Italy according to a children’s population-based study [3].

JBTS is a genetically heterogeneous disease with a major neurodevelopmental phenotype associated with a specific malformation of the cerebellum and brainstem, known as “Molar Tooth Sign” (MTS), and observed via neuroimaging [1]. Patients usually present with developmental delay (with possible intellectual disability), hypotonia, congenital cerebellar ataxia, and oculomotor apraxia [1,4]. These core features can be combined with secondary features involving organ defects, such as cystic kidney disease (and/or nephronophthisis), dystrophy of the retina (or coloboma), liver fibrosis, and dysmorphic phenotypes, such as polydactyly and orofacial abnormalities [1,5]. Apart from a case of an X-linked inheritance due to hemizygous variants in the *OFD1* gene, JBTS is a complex recessive genetic disease with more than 40 genes identified as causative genes; in approximately 60% of cases it is possible to establish a diagnosis through genetic screening [1].

The *CC2D2A* (OMIM *612013) gene is one of the most common causative genes of Joubert Syndrome-9 (JBTS9; OMIM #612285) [1]. The protein encoded by the *CC2D2A* gene (UniProt: Q9P2K1 … C2D2A_HUMAN) is located at the ciliary transition zone and its disruption is associated with the loss of ciliary protein localization [6]. Primary cilia are microtubule-based organelles important for the transduction of signals from outside a cell to inside a cell. They carry out this role by organizing and controlling the receptors and channels necessary for detecting these signals in their membrane domain. Disruption of proteins that participate in the sorting mechanism of the transition zone normally impairs the uptake of proteins, affecting the protein content of the ciliary compartment, and causing displacement of ciliary proteins’ normal location. Despite this, it is still poorly understood how the proteins perform this sorting process [6]. Studies have demonstrated that cilia have a crucial role in the neurodevelopmental aspect, and due to their ubiquitous localization, are also responsible for the pleiotropic clinical features of ciliopathies, such as JBTS9, that can overlap with Meckel syndrome because of clinical complexity and genetic heterogeneity [1]. In fact, *CC2D2A* variants can be responsible for 10% of Meckel syndrome cases, genotypes with CC2D2A hypomorphic variants may cause JBTS, and biallelic truncating variants can cause Meckel Syndrome [7].

In this report, we describe the identification via Whole Exome Sequencing (WES) of a novel missense variant c.3626C > T; p.(Pro1209Leu) and a novel 7.16 kb deletion inherited as a compound heterozygous in an Italian infant with typical JBTS9.

## 2. Materials and Methods

### 2.1. Family Trio

These results were obtained within the diagnostic activity of the Laboratory of Human Genetics regarding the nature of the genetic test. Informed consent was given for the genetic analysis of the family trio and possible consequences were discussed with the parents of our patient.

### 2.2. Whole Exome Sequencing

For most samples in our diagnostic laboratory, the WES workflow is commonly used as a wet part of sample preparation. Samples referred for exome analysis follow the WES pipeline; when processing data, we primarily filter the set of genes of interest related to the clinical indication. Genomic DNA of the patient was extracted from peripheral blood using the automatic extractor Symphony (Qiagen). WES was performed on genomic coding regions and exon–intron junctions (5 nucleotides) using the WES_v1: 20133 genes (SOPHiA Genetics) kit on a NovaSeq 6000 platform (Illumina—Italian Institute of Technology). The patient’s WES data underwent a comprehensive analysis of 37 genes (Appendix A) associated with Joubert Syndrome (JBTS) as suggested by the Human Phenotype Ontology browser (HPO). Data filtration and interpretation were carried out using SOPHiA DDM software, which included Copy Number Variations (CNV) analysis. The minimum target read depth was 20X with optimal coverage ≥ 98%. Genes marked with an asterisk (*) in Appendix A had lower coverage (≥82% and ≤97%) with a 20X target read depth; the *CC2D2A* gene had a coverage of 91.5%. Data were filtered for high quality (an alternative allele frequency ≥ 30% and rare variants with minor allele frequency, and MAF ≤ 0.5% according to the GnomAD database). The reference databases used were the human reference genome hg19 and The Human Gene Mutation Database dbSNP15. We used the Integrative Genomic Viewer (IGV) tool for the visualization of sequence data and variant calls. The pathogenicity of putative germline variants and residue conservation were evaluated according to the American College of Medical Genetics and Genomics (ACMG) [8] guidelines; bioinformatic analysis of novel variants used classifications from public databases (PolyPhen-2, SIFT, GERP, and others) supported by SOPHiA DDM, Varsome, and ClinVar.

### 2.3. Sanger Sequencing and Quantitative Polymerase Chain Reaction

Sanger sequencing was performed in the family trio for validation and segregation analysis of the missense variant (BigDye Cycle Sequencing Kit—Applied Biosystems), primers are available in Appendix A. The CNV detected using SOPHiA was validated via a quantitative polymerase chain reaction (qPCR) performed using a Light Cycler 480 (Roche). GAPDH was used as a reference gene for normalization. Each DNA sample was run in triplicate, in 20 µL of reaction mixture containing 25 ng of DNA, 0.10 mM of each primer, and 1x SYBR Green PCR MasterMix (Roche). The amplification conditions were as follows: a 5-min preincubation at 95 °C followed by 45 cycles of 10 s at 95 °C, 15 s at 60 °C, and 15 s at 72 °C. The PCR products were subjected to a linear temperature transition from 65 °C to 95 °C at 0.3 °C/s. LightCycler 480 Software was used for the analysis using the ΔΔCT method. Hemizygous deletion was determined when the relative copy number value for the specific sample, normalized to the reference sample, was less than 0.7. Primers used for qPCR are available in Appendix A.

## 3. Results

### 3.1. Clinical Description

We hereby describe the case of the only child of a healthy and non-consanguineous couple. At 20 weeks of gestation (WG), a routine morphology ultrasound revealed an abnormal rotation of the cerebellar vermis for which the mother was referred for fetal magnetic resonance imaging (MRI). The MRI showed mild dysmorphisms and hypoplasia of the cerebellar vermis. A female was born full term at 38 + 4 WG, from a natural birth; auxological parameters were: weight = 3110 g, length = 49 cm, head circumference = 34 cm, APGAR scores = 9 at minute 1 and 10 at minute 5. The patient never displayed abnormal respiratory patterns, such as apnea or hyperpnea. At 19 days old, she underwent a 3 Tesla brain MRI that revealed multiple abnormalities of the posterior fossa, including thickening, elongation, and horizontalization of the superior cerebellar peduncles, deep interpeduncular fossa, and vermal hypo-dysplasia (overall signs that result in the hallmark “MTS”) (Figure 1a). Furthermore, diffusion tensor imaging (DTI) revealed absent decussation of the superior cerebellar peduncles. The neuroradiological picture appeared compatible with a diagnosis of JBTS; therefore, the patient was directed to genetic testing at the Neurology Department. Neurological examination performed at 46 days of age revealed absent social smiling, very rare and fleeting eye contact, frequent tongue protrusion with rhythmic tongue motions, head rolling movements, mild truncal hypotonia, and two paramedian sacral dimples. An awake electroencephalogram (EEG) showed predominant theta rhythms and an absence of epileptic discharges. Despite not being observed during initial evaluations, nor reported by the parents, nystagmus was later noticed in subsequent ophthalmological examinations, especially in extreme side gaze. Furthermore, the infant displayed oculomotor apraxia with abnormal ocular pursuit movements and preferential fixation in extreme lateral positions. The LEA grating acuity test estimated a visual acuity of 1/15 in binocular vision. Cardiological evaluation and assessment of liver and renal functionality results were all within normal ranges. Ultrasound examination ruled out other anatomical malformations. Brainstem auditory evoked potentials (BAEPS) appeared to be normal. Neurological examination at 15 months old revealed a global developmental delay that required the initiation of physical and occupational therapy. Follow-up evaluations will be useful for further clinical characterization of the patient.

### 3.2. Genetic Findings

WES analysis identified in the *CC2D2A* (NM_001080522.2) gene a germline missense variant c.3626C > T; p.(Pro1209Leu) in exon 30. Allele frequency in the population showed that the single-base substitution is very rare (minor allele frequency, MAF = 0). Additionally, public databases were used to predict the impact of the variants in causing disease. The single nucleotide variant (SNV) was predicted as probably damaging via PolyPhen-2 and as a variant of unknown significance (VUS) by the ACMG [8]. Conservation analysis of nucleotides classified the variant as an evolutionarily conserved residue (genomic evolutionary rate profiling, GERP = 5.2). Sanger sequencing showed the SNV in a homozygous state, which was not expected considering the absence of frequency data in the global population (Figure 1b). Segregation analysis showed that the SNV was inherited from the father, who carried the variant in a heterozygous form (Figure 1c). A second finding was detected using WES and visualized using CNV analysis. A deletion of 7163 bases (NC_000004.11:g.15570905_15578067del) was identified, which generated a deletion of exons 28, 29, 30, and partial intron 30 in the *CC2D2A* gene. The first breakpoint is located nine bases upstream of the start codon of exon 28 (g.15570905), and the endpoint is in the middle of intron 30 (g.15578067). Segregation analysis (Figure 1c) and qPCR confirmed the inheritance of the CNV from the mother. The identification of the deletion encompassing the same exon of the SNV (Figure 2) explains the initial observation of the missense variant in a homozygous state when observed via Sanger sequencing, when actually it was in a hemizygous state. The two novel variants were submitted to the Leiden Open Variation Database (LOVD) with the individual ID #00431162 (https://databases.lovd.nl/shared/individuals/00431162, accessed on 6 February 2023), deletion DB-ID: CC2D2A_000257 and missense DB-ID: CC2D2A_000256.

## 4. Discussion

The *CC2D2A* gene, on chromosome 4, consists of 38 exons and encodes the protein named “Coiled-coil and C2 domain-containing protein 2A”, which has been proposed to have at least three coiled-coil domains for interaction with other proteins, a calcium-binding C2 domain crucial for vesicle fusion and trafficking, and a C-terminal region [1,9]. The protein encoded by the *CC2D2A* gene is part of a complex of proteins that localize in the transition zone of the cilia, between the ciliary axoneme and the basal body, where it interacts with proteins and takes part in cilia assembly [5].

The patient described here was referred for genetic testing due to a clinical picture compatible with JBTS (particularly MTS). Exome analysis identified two novel variants, a SNV and a CNV in the *CC2D2A* gene, which confirmed a JBTS9 diagnosis. Since the last follow-up, the patient has shown the presence of a global developmental delay, nystagmus, mild hypotonia, and oculomotor apraxia, all consistent with typical JBTS9 features. According to a recent cohort study [1], *CC2D2A* variants seem to be associated with a lower incidence of renal, retinal, and hepatic disease, and to have a lower impact on the development of the nervous system. Follow-up of our patient will be performed to evaluate the progression of the disease and possible complications.

Regarding our genetic findings, the germline missense variant inherited from the father localizes in exon 30, and the patient shows a hemizygous state due to a deletion of exons 28, 29, and 30 and part of intron 30 on the other allele inherited from the mother. The patient’s parents are both carriers and not affected; therefore, the novel SNV and CNV, together, are the most likely cause of JBTS9 disease based on a recessive inheritance pattern. Similar JBTS cases have been reported in the literature with a SNV and a loss of heterozygosity in the *CC2D2A* gene, but in different exons. In 2022, Ling-Xi Huang [10] identified fetuses prenatally diagnosed with JBTS. Among these, one affected fetus carried a splicing variant in intron 16, inherited from the mother (c.2003 + 2T > C), and deletion of exons 20 and 21, inherited from the father. In 2015, Yanhua Su [11] reported a fetus and a child both affected by JBTS carrying a missense variant (c.2999A > T) in exon 23 and a deletion of exon 20 and 21 in the other allele. We can assume that our report describes, for the first time, JBTS caused by two novel germline inherited variants, a missense and a deletion variant belonging to the same locus of the *CC2D2A* gene.

Moreover, exons 28 and 29 correspond to part of the C2 domain (from amino acid 1025 to 1203, UniProt: Q9P2K1) and exon 30 of both the C2 domain and the C-terminal region in the protein. The novel deletion described here has a total length of 7163 bases and comprises 125 amino acids (from amino acid 1133 to 1257, UniProt: Q9P2K1). The C2 domain has a length of 178 amino acids; the deletion removes 70 of those amino acids, meaning that approximately half of the amino acids are missing, and only 108 remain. Therefore, the CNV might have a pathogenic impact on the functional protein. The importance of the C2 domain is highlighted in the literature and is believed to participate in membrane-targeting activities such as subcellular localization and calcium-dependent phospholipid binding [1]. On the other hand, the C-terminal region does not have a clear function reported in the literature, but that it is highly conserved across species and studies indicates that it may have a role in cell polarity [1].

In conclusion, we identified two novel variants that will help expand the genetic spectrum of the *CC2D2A* variants causative of JBTS and better distinguish these genetic alterations from other ciliopathies. Further studies of RNA will be required to establish the length of the mutated transcript.

## Figures and Tables

**Figure 1 genes-14-00810-f001:**
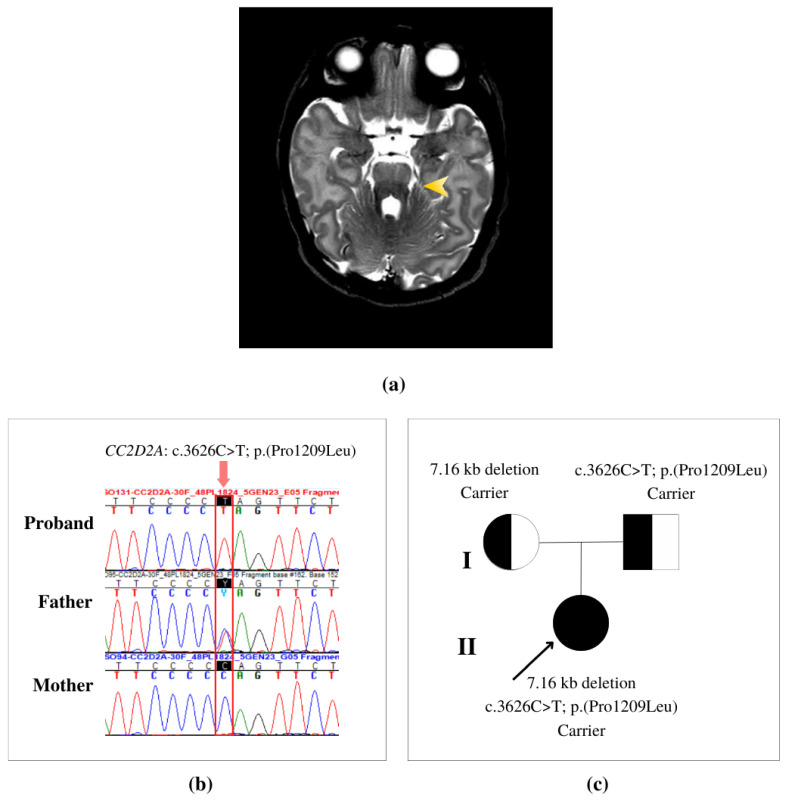
Clinical and genetic data. (**a**) Axial T2-weighted magnetic resonance imaging (MRI) shows thickening, elongation and horizontalization of the superior cerebellar peduncles, deep interpeduncular fossa, and vermal hypo-dysplasia resulting in the classic “Molar Tooth Sign” (yellow arrow); (**b**) validation via Sanger sequencing that the missense variant in the patient’s DNA is in a hemizygous state for the variant allele (T) because of the deletion of exon 30 on the other allele. The father shows the heterozygous state with the reference allele (C) and the variant allele (T); (**c**) pedigree: in the first generation (I) half-filled diagrams are carriers and in the second generation (II) the filled-in diagram is the affected patient.

**Figure 2 genes-14-00810-f002:**
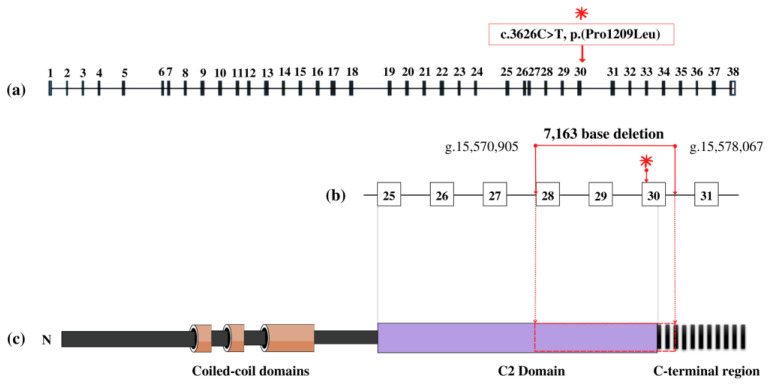
Schematic representation of the localization of the missense variant (shown by *) and 7163 base deletion in the *CC2D2A* gene and protein domains. (**a**) Exons (black boxes 1–38) and introns; (**b**) the enlarged view shows deletion within the genomic breakpoints, and the position of the missense in exon 30; (**c**) protein domains: coiled-coil domains, C2 domain, and C-terminal region.

## Data Availability

On reasonable request, the corresponding author will provide information supporting the study’s observations.

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
