# Peer review of "Exome Analysis Reveals Novel Missense and Deletion Variants in the CC2D2A Gene as Causative of Joubert Syndrome"

_genes, 2023, doi:10.3390/genes14040810_

Round 1

Reviewer 1 Report

In this manuscript, the authors report a clinical case of Joubert Syndrome, caused by novel mutations in the CC2D2A gene. The findings are relevant as they contribute our understanding of the etiology of ciliopathies.

I believe the manuscript should be better if some minor issues are addressed.

Minor issues:

-        The manuscript should undergo an extensive revision to correct the English. For example, the authors say “the mother was addressed”, but what I believe they mean is that the mother was directed to, or advised to do MRI. Also, in the discussion (line 190) the authors say “Our deletion”, and this is not very correct. They should say something like “the deletion identified by us”.  

-        The authors should confirm that the gene they used as a reference for qPCR was GADPH and not GAPDH.

Author Response

The manuscript was changed in the minor issues cited above.

Reviewer 2 Report

In this manuscript entitled “Exome analysis reveals novel missense and deletion variants in the CC2D2A gene as causative of Joubert Syndrome”, authors disclose that two novel variants of CC2D2A gene related the JBTS with clinical detections and WES method. Although, CC2D2A gene has been widely discussed and interpreted in other organisms and clinical cases, and previous works have been demonstrated that CC2D2A is the causative gene for ciliogenesis and ciliopathy, the new two variants about CC2D2A in this text expand our understandings of genetic spectrum of CC2D2A, but this work does not pay attention to and discuss the different clinical traits and cellular changes among the reported variants. If these limitations are overcome, the quality of this manuscript will reach a new level.

In general, there are still some concerned issues for careful consideration to make this manuscript more readable, clarity and credible.

1. In Materials and Methods section, line 60-61, the analysis method indicates that the WES data from the patient just compared with the known JBTS genes. In that way, would we miss any novel gene related to the JBTS?

2. In Fig 1, the MRI image shows the cerebellar molar tooth sign of JBTS. Is it better that adding the same location image shows the wild-type cerebellar clinical features? Also, it is better to use some symbols (e.g., arrow) in the figure to indicate the locations of explanations, making it more readable.

3. The figure (Fig 2) should be first addressed and labeled in the Results section, not first appeared in the Discussion section.

4. CC2D2A gene encodes the protein CC2D2A, not C2D2A. Are there some writing mistakes in the first paragraph of Discussion?

5. For further investigation, is it possible and applicable that check the cellular cilia from the patient cells, such as the ratio of ciliation and ciliary length.

6. Giving the readers a more readable list (or a table) for the reported CC2D2A variants in the literature and novel discovery in this manuscript is a clear presentation.

7. In the future studies, it is highly interesting to investigate those differences in clinical and cytopathic changes caused by these variants of CC2D2A gene and functional domains for the specific roles in cellular process. The more clinical and cytological data about CC2D2A we obtained, the clearer the molecular functions of this gene will be. As this CC2D2A is too long, if various functional complement fragments could be found for different variants, it may be very helpful for potential gene therapy.

Round 2

Reviewer 2 Report

This revised brief report and the response letter have addressed my scientific concerns. Thus, this revised version of manuscript is better organized and readable for describing two novel variants of CC2D2A gene from clinical and genetic data. Considering this type is brief report, it may be acceptable with  less experimental data.

In conclusion, this final manuscript is acceptable in the present form.

Author Response

Thank you for your comment.